# Effects of Litter Size and Parity Number on Mammary Secretions Including, Insulin-Like Growth Factor-1, Immunoglobulin G and Vitamin A of Black Bengal, Saanen and Their Crossbred Goats in Thailand

**DOI:** 10.3390/vetsci8060095

**Published:** 2021-05-31

**Authors:** Chollada Buranakarl, Sumpun Thammacharoen, Sapon Semsirmboon, Saikaew Sutayatram, Morakot Nuntapaitoon, Thasinus Dissayabutra, Kazuo Katoh

**Affiliations:** 1Department of Physiology, Faculty of Veterinary Science, Chulalongkorn University, Pathumwan, Bangkok 10330, Thailand; sprueksagorn@hotmail.com (S.T.); sapon_macmac@hotmail.co.th (S.S.); saikaew64@hotmail.com (S.S.); 2Department of Obstetrics, Gynaecology and Reproduction, Faculty of Veterinary Science, Chulalongkorn University, Pathumwan, Bangkok 10330, Thailand; morakot.n@chula.ac.th; 3Swine Reproduction Research Unit, Chulalongkorn University, Pathumwan, Bangkok 10330, Thailand; 4Department of Biochemistry, Faculty of Medicine, Chulalongkorn University, Pathumwan, Bangkok 10330, Thailand; thasinas@chula.md; 5Division of Functional and Developmental Science of Livestock Production, Graduate School of Agriculture, Tohoku University, Sendai 981-0845, Japan; kazuokatoh0@gmail.com

**Keywords:** colostrum, insulin-like growth factor 1 (IGF-1), immunoglobulin G (IgG), vitamin A (Vit A), litter size, parity

## Abstract

The present study aims to investigate the composition including concentrations of IGF-1, IgG and Vit A in colostrum and their effects by litter size and goat parity in 3 groups of goats; Black Bengal (BB), Saanen (SA) and their crossbred (BBSA). Thirty-eight goats were used (23 BB, 7 BBSA and 8 SA). The composition (fat, protein, lactose and total solid; TS) in colostrum (Day 0; D0) and milk (Day 4; D4 and Day 7; D7) were measured. The IGF-1, IgG concentrations were analysed in some samples collected at D0, D4 and D7 while Vit A was analysed only in colostrum. The results showed that colostrum components were similar among experimental groups. However, the colostral IGF-1 concentration of BBSA (983.0 ± 163.6 ng/mL) was higher than that of BB (340.7 ± 85.5 ng/mL, *p* < 0.01) and SA (417.1 ± 93.9 ng/mL, *p* < 0.01). The colostral IgG concentration of BB (8.2 ± 0.9 mg/mL) was lower than that of BBSA (12.9 ± 1.7 mg/mL, *p* < 0.05) and SA (12.9 ± 1.0 mg/mL, *p* < 0.01). Colostral Vit A concentration in BBSA (787.2 ± 152.6 µg/100 gm) was higher than that in BB (388.9 ± 84.3 µg/100 gm, *p* < 0.05) but was not different from SA (522.8 ± 96.9 µg/100 gm). Colostrum from all groups contained higher protein and TS but was lower in lactose concentration than milk. The IGF-1 and IgG concentrations in colostrum were much higher than in milk both D4 and D7 (*p* < 0.001). Additionally, litter size had no effects on colostrum contents but colostrum from goats with a higher parity number had higher IgG concentration. It is concluded that colostrum from BBSA may be superior when fed to BB newborn goats in terms of higher IGF-1, IgG and Vit A contents. Moreover, colostrum from goats with a high parity number contained more IgG content.

## 1. Introduction

Colostrum has a crucial role for newborn kids because it contains not only high concentrations of nutrients but also other important factors such as insulin-like growth factor 1 (IGF-1), immunoglobulin G (IgG) and vitamin A (Vit A) to accelerate kid growth and the immune defence mechanism. The somatotropin-IGF axis consists of growth hormone (GH), IGFs and their cellular receptors, which play a key role in systemic growth and development [1], while IgG is an immunological substance essential for the immune defence of kids and their growth development. A high concentration of IgG was found in the first few days and dramatically declined within 24 h after parturition in both monogastric and ruminant animals [2,3]. Lastly, Vit A is also related to growth and the immune system. Vitamin A deficiency causes defects of bone and maturation, loss of vision, reproductive disorders, and defects of the epithelial linings of the gastrointestinal (GI), reproductive, urinary and respiratory tracts leading to higher susceptibility to infection. The IGF-1, IgG and Vit A concentrations are present in high concentrations in colostrum and consumption by newborn kids plays a crucial role in term of growth, immunity and survival rate. Many factors have influences on IGF-1 and IgG levels both in colostrum and milk such as breed, nutrition and management [4,5,6,7]. In addition, few studies were investigated for the effect of litter size and parity number on colostrum composition and IgG content [2,8]. Nevertheless, the levels of IGF-1 and Vit A affected by litter size and parity number have not yet been elucidated in goats raised in tropical areas.

Black Bengal (BB) goats, a small-sized meat production goat, are raised well in tropical areas with hot and humid conditions such as in south-east Asia since they produce many pups and resist well to environmental heat and diseases. The average litter size is 2.13 per litter [9]. High litter size in BB can be problematic because some newborn kids are deprived of colostrum and milk. It is reported that goats giving birth to increased kid weight and had a high milk yield could improve pre-weaning survivability [10]. The source of colostrum and milk from the same species is preferable due to sufficient nutrients and the easy transfer of immunoglobulin. An attempt was made to produce a crossbred goat that was tolerant well against harsh environments and showed a better performance in both weight and milk yield [11,12]. The farm belonging to Chaipattana Foundation’s Bengal Goat Domestication Project is a farm that has the main purpose to conserve pure breed BB. However, BB newborns in high litter size were deprived of colostrum resulting in a low survival rate. To solve this problem, the dairy-type goat, SA had been imported into this experimental farm. The SA was also bred with buck BB to produce 50% crossbred (BBSA) offspring which had a high milk yield. Colostrum and milk that were obtained from SA and BBSA goats were kept and used to supplement BB kids that were born from high litter size.

Nevertheless, knowledge regarding the differences in colostrum and milk compositions including growth factor and immunological composition such as IGF-1, IgG and Vit A in BB, SA and their crossbred and the effect of litter size and parity number of goats have not been elucidated. The objectives of this study were to (1) determine the composition and levels of IGF-1, IgG, and Vit A in mammary secretions of BB, BBSA and SA, (2) determine the effect of litter size and parity on colostrum characteristics.

## 2. Materials and Methods

The procedures of this experiment were carried out in accordance with the guidelines and were approved by the Animals Care and Use Committee, Faculty of Veterinary Science, Chulalongkorn University (Animal Use Protocol No. 1831051).

### 2.1. Animals, Housing and General Management

The present study was performed on goats coming from the Chaipattana Foundation’s Bengal Goat Domestication Project, Chiang Rai province in the north of Thailand from June 2018–June 2019. Thirty-eight nursing female goats (two pure breeds and one crossbred defined as three experimental groups) were used for this study: (1) 100% Black Bengal (BB) (*n* = 23), (2) crossbred from 100% Black Bengal buck with 100% Saanen doe (BBSA) (*n* = 7) and (3) 100% Saanen (SA) (*n* = 8). The number of BB in this study was higher than BBSA and SA as goats in this project are mainly BB bred for genetic conservation. We have a few SA that used to breed with BB to produce a high milk yield for newborn BB goats. The number of BBSA crossbred is also limited since most of the mature goats will be distributed to the farmer at approximately 1–2 years of age. The age ranges of the goats were 23–117, 21–40 and 31–43 months old in BB, BBSA and SA respectively. All groups were housed in a separately conventional open-housing system while experimental groups were raised in different boxes. In general, the goats were naturally mated with selected bucks. The temperature and relative humidity inside the barn ranged from 14.6–32.8 °C (mean ± SD = 26.1 ± 4.5) and 42–91% (mean ± SD = 70.5 ± 13.1), respectively. The routine diet was based on concentrate (approximately 200 g/head/day) whose composition is shown in Table 1 and grasses at the experimental station. The Napier grass and Pangola hay were cut and supplied as roughage sources (approximately 2 and 1 kg/head/day, respectively). Goats were fed twice daily at 07:00 and 18:00 with free access to water. Grazing was allowed for animals for three to four hours in the morning throughout the year except that the time was limited during the rainy season. Both the does and bucks were vaccinated against foot-and-mouth disease virus (FMDV) while routine blood collection was performed for negative results to caprine arthritis encephalitis and brucellosis. The endoparasites and ectoparasites were monitored and controlled regularly using anthelmintic medication and dipping/spraying with pesticide, respectively. Goats in an advanced stage of pregnancy were kept in the pens for proper care during and after parturition. The pregnant goats received the diet regimen. The delivery of the goats was natural. All neonate goats received colostrum immediately after birth and remained with the mother for free milk consumption until weaning at 13 weeks of age. Some kids that were born from goats with a high litter size, especially the BB breed, were fed additional colostrum from other goats of any breeds. The signalment, body weight, vaccination programs, health status, parity number, current litter size and date of parturition were recorded for each goat.

### 2.2. Colostrum and Milk Collections

The colostrum and milk from each goat were collected in approximately 15–20 mL and 50 mL, respectively, by hand milking according to standard procedures and put into disposable plastic tubes. Colostrum was obtained on delivery day (D0) within 3 h after parturition (D0) while milk was obtained between 8:00–10:00 am on day 4 (D4) and day 7 (D7) after parturition. Immediately after collection, the sample tubes were placed in a −20 °C freezer until the day of analysis. Nutrient compositions were analysed in all samples. Analysis of IGF-1, IgG and Vit A-concentrations was performed in 9 BB, 6 BBSA and 6 SA, while Vit A was analysed only from D0 in 14 BB, 7 BBSA and 8 SA, respectively.

### 2.3. Analytical Procedure

#### 2.3.1. Colostrum and Milk Compositions

The colostrum and milk samples were thawed and kept in a water bath at 40 °C for 20 min before analysis of the microchemical composition including fat, protein, lactose and total solids (TS). The 3-folded dilution of colostrum with distilled water was performed before analysis. The approximately 30 mL of samples were sucked into the machine and analysed with infrared spectroscopy (MilkoScan FT2 instrument, Foss Milkoscan, Hillerød, Denmark).

#### 2.3.2. IGF-1 and IgG Concentrations

After the sample were thawed, they were pretreated with acid extraction according to the manufacturer’s instructions. The concentrations of IGF-1 in both colostrum and milk were analysed using solid-phase enzyme-labelled chemiluminescent immunometric assay (Immulite 2000 IGF-1, Siemens-Healthcare GmbH, Erlangen, Germany). The concentrations of IgG in both the colostrum and milk samples were determined by a goat specific goat IgG ELISA kit (Cat. no. K3231053P, Koma Biotech Inc., Seoul, Korea).

#### 2.3.3. Vitamin A Concentration

The Vit A in colostrum was measured using high-performance liquid chromatography (HPLC) with a UV detector as described previously [13]. In brief, after the samples were thawed, the fat was saponified with alcoholic potassium hydroxide solution. The Vit A was extracted from unsaponifiable portions with n-hexane and evaporated under nitrogen. The residue was dissolved in methanol and injected into the reverse phase C18 µ-Bondapak (3.9 × 300 mm) column (Water Corp., Milford, MA, USA). The mobile phase consisted of methanol (RCI Labscan Ltd., Bangkok, Thailand) set at the flow rate of 1 mm/min and Vit A was detected at 340 nm using the HPLC system (HPLC-Shimadzu, Kyoto, Japan). The recovery of the added standard at the level 5–20 ppm was 99.5% and a standard deviation of 3.78.

### 2.4. Statistical Analyses

All statistical analyses were performed using SAS version 9.0 (SAS Institute Inc., Cary, NC, USA). The colostrum composition including IGF-1, IgG and Vit A and their effects from litter size and goat parity number were analysed using the general linear model procedure. The model included the fixed effects of goat group (BB, BBSA and SA), litter size (1 and 2) and goat parity numbers (1–3, 4–6 and >6).

The effects of goat group and time on colostrum and milk composition (i.e., fat, protein, lactose, TS, IGF-1 and IgG) were analysed by using a general linear mixed model procedure. The model included the fixed effects of goat group (BB, BBSA and SA), time (0, 4 and 7) and interaction between goat group and time. The following model was applied to analyse data:Y_ijk_ = m + B_i_ + T_j_ + BT_ij_ + S_k_ + O_ijk_
where Y_ijk_ is the response variable, m is the overall mean, B_i_ is the fixed effect of the goat group (i.e., BB, BBSA and SA), T_j_ is the fixed effect of time (i.e., 0, 4 and 7), BT_ij_ is the interaction between goat group and time, S_k_ is a random component related to the goat and O_ijk_ is the residual error component. The goat ID was included as a random effect.

The relationship among the colostrum compositions was determined by Pearson correlation. The data are presented as least–squared means or mean ± standard error. A probability value of *p* < 0.05 was regarded as being statistically significant.

## 3. Results

### 3.1. Goat’s Characteristics

The bodyweight of goats in BB (24.4 ± 1.1 kg) was significantly lower than BBSA (31.9 ± 2.5 kg) (*p* < 0.01) and SA (35.5 ± 1.3 kg) (*p* < 0.001). There was no difference in body weight between BBSA and SA groups. The goat’s age in BB, BBSA and SA was 53.6 ± 4.7, 37.4 ± 3.9 and 64.5 ± 8.2 months old, respectively, and was not different among groups. The goat parity number in BBSA was lower than BB (2.4 ± 0.3 vs. 5.3 ± 0.6) (*p* < 0.05) but not different from SA (4.4 ± 0.7). The litter sizes were 1.9 ± 0.1, 2.0 ± 0 and 1.5 ± 0.2 kids/litter in BB, BBSA and SA, respectively.

### 3.2. Colostrum Compositions, IGF-1, IgG and Vit A Concentrations among Groups

The compositions, IGF-1, IgG and Vit A concentrations in colostrum are shown in Table 2. There were no differences in concentrations of fat, protein, lactose and TS among groups. However, the IGF-1 concentration in BBSA was significantly higher than both BB and SA (*p* < 0.01). The IgG concentration of BB was significantly lower than both BBSA (*p* < 0.05) and SA (*p* < 0.01). The Vit A concentration in colostrum of BBSA was higher than BB (*p* < 0.05) but was not different from SA.

### 3.3. Relationships between Colostrum Composition

The relationships among colostrum compositions are shown in Table 3. Total solid correlated positively with fat and protein but negatively with lactose concentrations. Lactose also had a negative relationship with protein concentration. Vit A in colostrum was positively correlated with fat content. No relationship was found among IGF-1, IgG and Vit A in colostrum.

### 3.4. Effects of Litter Size on Colostrum Compositions and Levels of IGF-1, IgG and Vit A

No differences in fat, protein and lactose concentrations and TS in colostrum from goats that delivered singly compared with twins (Table 4). However, the lactose concentration in colostrum from the 2 litter size goats tended to be lower than 1 litter size goats (*p* = 0.051). Goats delivering single or twin kids produced colostrum with the same amount of IGF-1, IgG and Vit A.

### 3.5. Effects of Parity on Colostrum Composition and Levels of IGF-1, IgG and Vit A

Goat parity number did not affect colostrum fat, protein concentrations and TS (Table 5). Colostral lactose in parity >6 goats tended to be lower than in parity 1–3 goats (*p* = 0.088). The IGF-1 in colostrum of parity > 6 goats had a tendency to be higher than that of parity 1–3 goats (*p* = 0.080). The IgG concentration in colostrum of parity >6 goats was significantly higher than that of parity 1–3 goats (*p* < 0.05) while Vit A concentration in colostrum of parity >6 goats also tended to be higher than that of parity 1–3 goats (*p* = 0.129).

### 3.6. Transition from Colostrum to Milk

The effects of group and day and their interaction on colostrum and milk are shown in Table 6. The fat and protein contents were similar among groups. Lactose concentration in colostrum and milk in BBSA was higher than BB while TS in BB was the highest. The IGF-1 concentrations in BBSA were higher than both BB and SA (*p* < 0.001). The IgG concentration showed no differences among groups.

When considering all groups, the fat content was unchanged over time. The protein concentration and total solids were reduced while lactose concentration was increased during colostrum being transitioned to milk (*p* < 0.001). The IGF-1 and IgG concentrations were dramatically reduced by more than 90% and 98%, respectively since D4 (*p* < 0.001).

## 4. Discussion

Breeding between BB and other breeds was reported to improve both weight and milk production [12,14]. In general, colostrum and milk from both SA and BBSA could be supplied to newborn BB kids when mother BB delivered many kids with a limited amount of colostrum. The present study is the first report about the level of IGF-1, IgG and Vit A in colostrum and milk of BB, BBSA and SA and their effects by litter size and goat parity.

### 4.1. Colostrum Composition, IGF-1, IgG and Vit A Contents in BB, BBSA and SA Goats

Colostrum contains high concentrations of nutrients for growing to the next stage in young goats. The present study showed that colostrum fat, protein and lactose were not different among groups. According to the previous studies, the colostrum contents were reported earlier in many species [15,16]. The colostrum compositions in SA were comparable to SA in the previous studies except for variations in lactose and protein [17,18]. The TS is composed of mainly fat, and protein which is suggested by the presence of positive relationships between parameters. However, the negative relationship between TS and lactose suggests that lactose may be secreted from the mammary glands at a low level on the first day compared with other solids and lactose may not contribute to colostrum volume.

The BBSA crossbred was younger than the pure BB and SA goats while the difference in age can influence the composition of colostrum. Additionally, the colostrum yield that may affect the composition of colostrum was not measured in the present study due to newborn kids being raised with the mother and still consuming the colostrum.

This report is the first report regarding the IGF-1 concentration in colostrum of BB, SA and their crossbred. The colostral IGF-1 concentration of crossbred BBSA was higher than those of both BB and SA goats. Intake of colostrum and IGF-1 stimulates growth and development especially the maturation of the gastrointestinal tract of the neonate that was reviewed earlier [19]. It is interesting whether this high concentration of IGF-1 in colostrum is contributed to the high growth rate of kids in this crossbred.

The colostral IgG concentrations in BBSA and SA were higher than in BB. The level of IgG in SA in this study was approximately 12 mg/mL and slightly lower than previous reports in which SA had IgG concentration ranged from 4.8–75 mg/mL [8,17]. It is crucial to collect colostrum at the same time postpartum since the concentration may vary tremendously. Not only the time of collection but breed may be a prime factor for the difference in IgG levels. Reports of different goat breeds raised in the same environment and management had different levels of colostrum IgG have been demonstrated [20]. A study of IgG in many breeds of goats in Switzerland and Germany found the IgG levels varied between 4.8 and 75.0 mg/mL collected within 6.5 h postpartum [8]. When considering the type of goats, the highest IgG was found in the meat-type goat, Boer (61.0 ± 10.3 mg/mL) and the lowest was found in the milk-type goat, Bunte Deutsche Edelziege (26.5 ± 12.5 mg/mL) [8]. Nevertheless, the present study showed the milk type-goat, SA had higher IgG than the meat-type goat, BB. Therefore, breed rather than the type of goats had a strong influence on colostrum composition. For goats, Castro et al. (2007) [21] showed that feeding 4 g of IgG/kg of BW within 24 h provided passive immunization in approximately 80% of kids as measured by the kid’s plasma IgG levels. On our farm, the average birth weight of BB, BBSA and SA was 1.26, 2.42 and 2.53 kg, respectively. The newborn BB, BBSA and SA may likely need 615, 750 and 785 mL, respectively to accommodate these criteria. The colostrum yield in dairy goats was approximately 0.86 L/day [22]. From these data, it is difficult for BB goat to produce adequate colostrum for kids that were born from high litter size.

Vit A in colostrum of BBSA was the highest and was significantly higher compared with BB. Vit A concentrations in colostrum were reported earlier in many species [23]. High colostrum Vit A was reported in dairy cattle especially during the first week post-partum [24]. In ruminants, it is important to ensure that Vit A in colostrum will be high enough for feeding newborn kids with low reserves of Vit A and low plasma concentration [25]. Feedstuffs may contain low Vit A due to poor storage conditions, moisture, and heat while Vit A can be oxidized quickly by light. Therefore, the mother goats themselves may be in a Vit A deficient condition. Calves not receiving first-day colostrum had carotene, retinol and alpha-tocopherol lower than calves receiving the first-day colostrum [26]. In the present study, a relationship between Vit A and fat contents in colostrum was found as well-known, Vit A is fat-soluble. This relationship had been found earlier in the milk of sheep [27].

### 4.2. Effects of Litter Size on Composition, IGF-1, IgG and Vit A Contents

Litter size had no effects on the composition of colostrum except lactose tended to be lower from colostrum in goats that delivered twins. A report of colostrum from high litter size goats revealed lower lactose and protein concentrations [2]. The lower lactose concentration may be due to dilution effects during high milk production associated with high litter size or parity number which needs further study.

Our results showed no effect of litter size on IGF-1, IgG and Vit A contents on colostrum. This is the first report showing colostral IGF-1 and Vit A levels in relation to litter size and parity number in goats. No effect of litter size on colostral IgG contents similar to our study was found in Majorera goats [28], Murciano-Granadina goats [2] and goats of many breeds [8]. However, controversy surrounds some studies. In Hungarian white goats, the IgG contents in colostrum collected at 0.5–1.5 h after parturition in twin producing goats was higher than that with single progeny [29]. Studied of the colostral IgG contents in Damascus goats delivering singleton, twins and non-pregnant showed the highest IgG in goats with twins which was significantly higher than non-pregnant goats [30]. It is postulated that increased mammogenesis of dams in response to multiple births may contribute to higher IgG contents for adequate delivery to offspring. Unfortunately, the colostrum yield or colostrum IgG consumption was not measured and calculated.

### 4.3. Effects of Parity Number on Composition, IGF-1, IgG and Vit A Contents

Similar to litter size, parity number had no significant effects on composition as shown earlier in goats of many breeds [8] although lactose tended to be lower. However, both lower protein and lactose concentrations were found previously [2]. The IGF-1, IgG and Vit A contents were higher especially when the parity number was >6. This is the first report that showed the association of parity number of goats with colostral IGF-1 and Vit A contents in goats. In multiparous cows, colostral IGF-I concentration was higher than that of first-lactation heifers [31]. The controversy was found for Vit A. Vit A levels in colostrum and milk in primiparous cows were higher than those in multiparous cows [32]. The reason may be due to age in which the body store of Vit A may be depleted when animals are getting older. Nevertheless, dietary intake of Vit A had a strong influencing effect on hepatic Vit A storage as shown in calves fed milk replacers containing different concentrations of vitamin A [25]. Higher Vit A in colostrum in goats with older age or high parity number in the present study may reflect higher Vit A storage due to feeding higher quality roughage to goats in this farm over the past few years.

In the present study, higher parity goats produce colostrum with significant higher IgG content. Previous studies in goats showed no effect of parity on colostral IgG contents [2,28]. This is the first report in goats although higher IgG in colostrum has been found in Japanese black multiparous cows with increasing age [33]. Increased colostral IgG along with a high parity number may be related to older age in which animals could be exposed to a greater amount of pathogenic antigens in their lifetime. Whether changes in colostral IgG related to age or parity number need further study.

### 4.4. Transition from Colostrum to Milk

Changes from colostrum to milk within 7 days showed a decline in protein and TS with an elevation of lactose. Similar results were found in the colostrum and milk of many breeds such as SA [17], SA crossbred (Sannen x Beetal goat and Alpine x Beetal goats) [20], Murciano-Granadina goats [2] and Majorera goats [34,35]. Large differences in protein concentration between colostrum and milk reflect the immunoglobulin, which is important for passive immunity especially in ungulates [36]. Higher lactose in milk may contribute to higher milk yield since it is responsible for 50% of the osmotic pressure of milk, while its synthesis draws water into Golgi vesicles [37]. The fat content varied between being elevated [38] or declined [18,28] when colostrum was transitioned to milk.

The IGF-1 contents in both colostrum and milk in many species including goats has been reviewed and shows a much higher concentration in colostrum [39]. In cattle, colostrum contained 870 ng/mL of IGF-1 and declined to 150 ng/mL at 51–80 h postpartum [40] while another study reported a reduction in IGF-1 colostrum by about six times between 6 to 18 h after parturition [5]. One study showed IGF-1 reduction by 77.44% for buffaloes and 76.03% for cows after 5 days postpartum [41]. Our results in goats showed a reduction by 93% at 4 days postpartum, which was kept at the same levels at 7 days postpartum, showing that the trend of reduction in IGF-1 in goat colostrum was similar to cattle. A positive correlation between kid’s blood IGF-1 concentrations and some body trait measures was found in goats during the prepubertal period (up to 5 months old) [42]. However, another hormone such as thyroid hormone later had a positive effect on growth and mohair production of kids [43].

The colostral IgG in the present study was progressively declined by 98% on D4 and remained low at D7, as similarly shown by Yang et al. (2009) [17]. Similar changes from the colostrum to milk were found in many breeds such as Majorera goats [34,35], Sannen x Beetal goat and Alpine x Beetal goats [20] and Murciano-Granadina goats [2]. However, no available data were reported in the IgG levels in BB and their crossbred. Our results indicated that the level of IgG in milk reached lower as early as 4 days postpartum than that in buffaloes and cow that reached minimum values after 5 days [41]. The controversy, however, was reported in one study with SA colostrum, which showed that the IgG level was 1.88 mg/mL immediately at birth and then increased up to 14.0 mg/mL and 13.5 mg/mL at 24 and 48 h after birth, respectively [18]. Therefore, the timing of collection may influence the colostrum IgG level, particularly immediately after parturition.

Although this study measured Vit A-concentrations only in colostrum, the level in colostrum was higher than those reported in the milk of Aardi and Masri breeds (77 and 203 IU/100 gm, respectively) [44]. Vit A content in colostrum (2.55 mg/kg) was declined to 1.07 and 0.38 mg/kg within 24 and 48 h postpartum in Yak [45]. A slow declining degree of Vit A-concentrations by 18% and 49% was found in colostrum at 14 days after parturition in Egyptian buffaloes and Holstein cows, respectively [41].

## 5. Conclusions

Most of the composition of colostrum was similar, while concentrations of IGF-1, IgG and Vit A were different among groups. Colostrum of BBSA had higher IGF-1, IgG and Vit A than BB, and colostral IgG concentrations were greater in higher parity goats than lower parity goats. These findings suggested that BBSA colostrum is superior for feeding small size BB newborns to help their growth and immunity. Both IGF-1 and IgG concentrations in colostrum were dropped dramatically in all groups within 4 days of parturition, which indicates an essential role of colostrum consumption for newborn kids although it was basically similar to other breeds and animals.

## Figures and Tables

**Table 1 vetsci-08-00095-t001:** The chemical composition of diet.

Diet	Concentrate	Pangola Hay	Napier Grass
Dry matter (%)	92.01	92.51	19.62
Chemical composition, (g/100g DM)			
Ash	10.11	8.37	9.68
Organic matter	89.89	91.63	90.32
Crude protein	17.60	4.53	10.70
Crude fat	4.52	1.61	1.99
Crude fibre	14.80		
Neutral detergent fibre		71.82	69.93
Acid detergent fibre		45.61	46.89
Calcium	2.54	1.50	1.43
Phosphorus	0.66	0.26	0.25
Digestible energy, MJ/kg	11.13		
Metabolizable energy, MJ/kg	10.90		

DM: dry matter. MJ: Megajoule.

**Table 2 vetsci-08-00095-t002:** The colostrum composition in three groups of goats: mean ± standard error of the mean.

		Group	
Compositions	BB	BBSA	SA
Fat (g/100 g)	7.0 ± 0.6	7.4 ± 1.3	7.1 ± 1.0
Protein (g/100 g)	12.1 ± 0.6	11.0 ±1.3	12.2 ± 0.9
Lactose (g/100 g)	3.6 ± 0.1	3.8 ± 0.2	3.5 ± 0.1
TS (g/100 g)	23.4 ± 0.7	23.3 ± 1.5	23.6 ± 1.1
IGF-1 (ng/mL)	340.7 ± 85.5 ^b^	983.0 ± 163.6 ^a^	417.1 ± 93.9 ^b^
IgG (mg/mL)	8.2 ± 0.9 ^b^	12.9 ± 1.7 ^a^	12.9 ± 1.0 ^a^
Vit A (µg/100 g)	388.9 ± 84.3 ^b^	787.2 ± 152.6 ^a^	522.8 ± 96.9 ^ab^

^a,b^ Means within a row among groups with different superscripts differ (*p* < 0.05). BB: 100% Black Bengal. BBSA: 50% Black Bengal (Buck) with 50% Saanen (Doe). SA: 100% Saanen. TS: Total solids. IGF-1: Insulin-like growth factor-1. IgG: Immunoglobulin G. Vit A: Vitamin A.

**Table 3 vetsci-08-00095-t003:** Correlation coefficients of colostral compositions, IGF-1, IgG and Vit A in goats.

Compositions	Protein	Lactose	TS	IGF-1	IgG	Vit A
Fat	−0.287	−0.151	0.666 ***	0.159	0.105	0.498 **
Protein		−0.654 ***	0.512 **	0.0418	−0.0815	−0.361
Lactose			−0.569 ***	0.0696	−0.00771	0.0675
TS				0.178	0.00580	0.142
IGF-1					0.200	0.299
IgG						0.157

*p* < 0.05; ** *p* < 0.01; *** *p* < 0.001. TS: Total solids. IGF-1: Insulin like growth factor-1. IgG: Immunoglobulin G. Vit A: Vitamin A.

**Table 4 vetsci-08-00095-t004:** Effects of litter size on colostral compositions, IGF-1, IgG and Vit A concentrations: mean ± standard error of mean.

	Litter Size	
Compositions	1	2
Fat (g/100 g)	7.3 ± 1.0	7.1 ± 0.6
Protein (g/100 g)	11.2 ± 0.9	12.4 ± 0.6
Lactose (g/100 g)	3.7 ± 0.1	3.5 ± 0.1
TS (g/100 g)	23.1 ± 1.1	23.9 ± 0.7
IGF-1 (ng/mL)	639.5 ± 113.6	521.1 ± 63.1
IgG (mg/mL)	11.7 ± 1.2	11.0 ± 0.6
Vit A (µg/100 g)	558.0 ± 115.7	574.6 ± 65.4

TS: Total solids. IGF-1: Insulin-like growth factor-1. IgG: Immunoglobulin G. Vit A: Vitamin A.

**Table 5 vetsci-08-00095-t005:** Effects of goat parity number on colostral compositions, IGF-1, IgG and Vit A concentrations: mean ± standard error of mean.

	Parity		
Composition	1–3	4–6	>6
Fat (g/100 g)	6.4 ± 0.7	7.1 ± 0.9	8.1 ± 1.3
Protein (g/100 g)	11.4 ± 0.7	12.1 ± 0.8	11.8 ± 1.2
Lactose (g/100 g)	3.7 ± 0.	3.7 ± 0.1	3.4 ± 0.2
TS (g/100 g)	22.3 ± 0.8	23.8 ± 1.0	24.3 ± 1.4
IGF-1 (ng/mL)	403.9 ± 92.1	586.7 ± 108.8	750.2 ± 139.0
IgG (mg/mL)	8.8 ± 0.9 ^b^	11.5 ± 1.1 ^ab^	13.9 ± 1.4 ^a^
Vit A (µg/100 g)	530.8 ± 91.1	571.2 ± 102.3	696.9 ± 136.3

^a,b^ Means within a row among groups with different superscripts differ (*p* < 0.05). TS: Total solids. IGF-1: Insulin-like growth factor-1. IgG: Immunoglobulin G. Vit A: Vitamin A.

**Table 6 vetsci-08-00095-t006:** Compositions of colostrum and milk among groups and days.

	Group				Day				*p*-Value		
Composition	BB	BBSA	SA	SEM	0	4	7	SEM	Group	Day	Group X Day
Fat (g/100 g)	7.3	5.8	6.2	0.7	6.7	6.3	6.3	0.6	0.167	0.738	0.517
Protein (g/100 g)	9.3	7.5	7.8	0.7	12.0 ^x^	7.1 ^y^	5.5 ^y^	0.7	0.095	<0.001	0.833
Lactose (g/100 g)	3.9 ^b^	4.2 ^a^	4.0 ^ab^	0.1	3.6 ^y^	4.1 ^x^	4.3 ^x^	0.1	0.088	<0.001	0.844
TS (g/100 g)	21.5 ^a^	18.6 ^b^	19.2 ^b^	0.9	23.3 ^x^	18.7 ^y^	17.3 ^y^	0.9	0.024	<0.001	0.452
IGF-1 (ng/mL)	177.4 ^a^	268.2 ^b^	139.7 ^a^	34.2	515.8 ^x^	37.3 ^y^	32.3 ^y^	32.4	0.028	<0.001	0.004
IgG (mg/mL)	3.522	3.41	4.25	0.40	10.75 ^x^	0.24 ^y^	0.18 ^y^	0.38	0.262	<0.001	0.092

^a,b^ Means within a row among groups with different superscripts differ (*p* < 0.05); ^x,y^ Means within a row among days with different superscripts differ (*p* < 0.001). BB: 100% Black Bengal. BBSA: 50% Black Bengal (Buck) with 50% Saanen (Doe). SA: 100% Saanen. SEM: standard error of mean. TS: Total solids. IGF-1: Insulin-like growth factor-1. IgG: Immunoglobulin G.

## Data Availability

The data presented in this study are available within the article.

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
