# Peer review of "Effects of Litter Size and Parity Number on Mammary Secretions Including, Insulin-Like Growth Factor-1, Immunoglobulin G and Vitamin A of Black Bengal, Saanen and Their Crossbred Goats in Thailand"

_vetsci, 2021, doi:10.3390/vetsci8060095_

Round 1

Reviewer 1 Report

Problems found:

1) The number of births between breeds was not reported and it can influence the composition of colostrum, impairing the conclusions. In the statistical analysis it was informed that the number of parity was included in the model, but it is necessary to inform the values of the different breeds.2) It would be interesting to inform the amount of colostrum produced by the different breeds, since the amount produced can influence the composition.3) in line 169 it was informed that the average age of the goats was: 53.6 37.4 and 64.5 YEARS, this is not possible! It’s important to highlight that different ages can influence the composition of colostrum ...4) The number of goats per treatment is very different, almost 200% ! 

Author Response

Answer

  1. The litter size and parity number of each breeds were shown in results section under the topic of “Doe’s characteristic”.
  2. Unfortunately, we did not collect colostrum from each doe since the kids were allowed to stay with mother goats and consumed colostrum freely. I put this issue in limitation section. The data on colostrum yield in BB and SA are limited and unavailable. In crossbred, nobody has been studied yet except the body weight conformation in which weight of crossbred F1 is similar to SA (Nuntapaitoon et al., Growth performance of Black Bengal, Saanen and their crossbred F1 as affected by sex, litter size and season of kidding. Anim Sci J., In press).
  3. I changed “years old” to “months old”. I also put in limitation that the age of goats may influence the composition of colostrum.
  4. The number of goats per treatment is small and very different due to limit number of pure SA and crossbred goats in farm.

Reviewer 2 Report

The paper vetsci-1198444, entitled “Effects of litter size and parity number on colostrum compositions, Insulin-Like Growth Factor-1, Immunoglobulin G and Vitamin A of Black Bengal, Saanen and Their Crossbred in Thailand”, submitted as article, deals with some dairy characters of different genetic types in goats raised in harsh environments, as far as I understand. The topic could be potentially interesting, but the paper cannot be considered because the very poor-quality English and the extremely low quality of scientific writing.

For instance, please note:

- the aim of the paper is unclear

- the crossbred BBSA should not be considered a breed

- experimental conditions should be better described

- can the Author confirm that kids were weaned at 13 months of age?

- the statistical analysis should consider that processed data are repeated measures on same subjects

- does’ age (l. 168,  months not years!) ) should be reported in Material and methods section

- Are tables 2 and 4 reporting results from processed data or descriptive statistics?

Although the topic falls within the scope of the Journal, the paper needs great improvements to be re-submitted for publication.

Author Response

Answer

  • The paper was sent to improve the English proficiency.
  • The aim of this paper was rewritten as we focus only on colostrum, not in milk.
  • The crossbred BBSA was not a specific breed but it is easier for reader to know that we have 3 different groups with 3 different genotypes. We added in M&M that we studied in 2 pure breeds and 1 crossbred defined as three breed groups
  • We added that all breeds were house separately and added the age range of each breeds.
  • Kids were weaned approximately 13 weeks of age and separate from mother.
  • The statistical analysis in Table 6 using the mixed model procedure of SAS. The testing is repeated measure ANOVA.
  • The doe age was changed from “years” to “months”. The age ranges for each group were added in materials and methods
  • The results from Table 2, 4 and 5 came from the processing data already, not from the descriptive data. We added the test in the title of Tables.

Reviewer 3 Report

A) General comments

In this straightforward descriptive study, the authors address important knowledge gaps by comparing the levels of important components in the milk in different goat breeds and determine the impacts of doe parity, litter size and time since delivery. The solid rationale provided by the literature review, and the robust statistical analyses are strengths. Enthusiasm is mildly dampened by unclear sample sizes, rough grammar and numerous typographical errors.

B) Specific comments

  1. Line 90: should sample size be listed as 38 or 35?
  2. Table 1: if the listed values are all reported as % dry matter, what are the units for dry matter in line 1? If it was normalized to itself, shouldn’t it be 100%?
  3. Line 169: do you mean months-old rather than years-old?
  4. Table 4: what are the N values? Are there sufficient singleton litters for well-powered statistical analysis or should that be listed as a limitation of the study?
  5. Spellchecker should be re-run to catch typographical errors, e.g. “in in” on line 269 and “taht” on line 350.

Author Response

Answer

  1. Line 90: sample size is 38.
  2. We inserted the “chemical composition, %DM” to explain that the lower compositions are normalized with DM as percent. It should be noted here that when “organic matter” and “Ash” are combined, it’s always 100%. The rest of composition does not need to meet 100%.
  3. Yes, we changed to months-old already.
  4. The N values of singleton for colostrum composition was 10 and for twin was 25. For IGF-1 and IgG, they were 6 and 14 and for Vit A, they were 7 and 21, respectively. I also added the limitation of this study that the numbers of goats in this study for BBSA and SA were limited.
  5. The typing errors were corrected, and the MS was sent for proof editing again.

Reviewer 4 Report

The present work is a worthy and valuable study. It invokes an applied topic in goat husbandry, trying to evoke solutions concerning productivity and surviving offspring of BB goat. These matters have been developed through investigation on colostrum composition and IGF-1, IgG and Vit A contents in BB, SA and their crossbred goats.

The document is well presented both on writing style and structuring that only minor error can be found.

Introduction: It includes general information about colostrum contents in different species and goat in particular. However, more details are needed to emphasise its role for the living and growth of newborn goats, particularly concerning the three parameters quoted in this study: IGF-1, IgG and Vit A. The objectives were lucid and accurate.
Material & Methods: The study was performed on 38 nursing females goats belonging to three breeds (BB: 20, BBSA: 7, SA: 8). My remarks are: why the proportions of each breed are different? And also the three breed were housed together or separated? It should be clarified.
The protocol is well put together and the methods used for quantifying different parameters including the statistical analysis are all valid.

Results: In general, the results obtained are accurate and of good scientific value that could be used to improve performances of goat farming. They were presented in tables and were well commented. However, I think that females with single parity should be excluded from the compared groups.
Lines 168-169: The given animal ages should be rectified. 

Discussion: Generally, the obtained results have been discussed and compared to those of other related studies. It includes abundant references and related data in other species. It was conducted according to specific results obtained; while, an overall discussion was omitted. 
Conclusions: are concise and reflect the obtained results and predefined objectives.
(See the joined manuscript version including other minor comments)

Author Response

Answer

Introduction - We added a sentence regarding the important of colostrum in newborn.

Materials and methods - The study was performed in three breeds (BB: 23, BBSA: 7, SA: 8). The number is different according to the project farm that conserve black Bengal. We had many BB in farm, but we have only a few SA and the crossbred BBSA was also limited. Most of BBSA was distributed to farmer after 1-2 years old. So, the age of BBSA in this study is also younger than other breeds. I put this in limitation of the study. We put on materials and methods that the three breeds were raised in separated pen.

Results - If we excluded the female with single parity, the trend of results is the same. IgG in 2-3, 4-6 >6 was 8.8 ± 1.2, 11.6 ± 1.2 and 14 ± 1.5 mg/dl, respectively and the significant differences of IgG are also the same. Therefore, we prefer to include single parity that the number of samples are higher.

We change the age from years old to months old already.

Discussion - The detail discussion was written according to results which was measured only in colostrum but not the outcome in newborn kids. The data may be basic data that we speculate of the usefulness of colostrum from SA and BBSA to feed BB newborns. However, the growth or survival rate of newborn was not shown in this study. We added some more discussion as suggestion.

Reviewer 5 Report

I have some comments and suggestions to your manuscript:
-    In title, I recommend to include the species of animal (goats in this case).
-    Lines 24: Black Bengal is missing before BB. I recommend to use goat instead doe. Use it in hole body of manuscript.
-    Line 90: Check the number on goats. In Materials and methods 38 goats were used but in abstract 35 were used. According with the number of goats in every treatment, the correct number is 35. 
-    Why was the number of animals in each treatment not homogeneous?
-    Could you explain why collections of colostrum and milk were not perform in all experimental goats?

Regards,

Author Response

Answer

  • We added “goats” in title.
  • We added “Black Bengal” before abbreviation. We changed Doe to goat as your suggestion.
  • It is 38. I already corrected in abstract.
  • The number of animals in each treatment not homogeneous due to the limitation of the project that conserve only BB. We only had a few SA. I added this issue in the limitation of the study in manuscript. Please see below.

Limitations

The number of goats in each group was different with BB was the highest. Since, goats in this project are mainly BB breed for genetic conservation. We have a few SA that used to breed with BB in order to produce high milk yield for newborn BB goats. The number of BBSA crossbred is also limited since most of the mature goats will be distributed out for the farmer at approximate 1-2 years of age. Therefore, the numbers of goats in this study for BBSA and SA were limited. The BBSA crossbred was younger than the pure BB and SA goats while the different of age can influence the composition of colostrum. Additionally, the colostrum yield that may affect the composition of colostrum was not measured in the present study due to newborn kids were raised with mother and still consume the colostrum.

  • The milk samples especially on D4 and D7 could not be collected in some goats primarily due to the limit amount of milk to feed the newborn kids especially when doe delivered many kids in that litter size. This farm prefers to use goat colostrum and milk for feeding the first month old newborns rather than using the bovine milk replacer.

Round 2

Reviewer 2 Report

The paper vetsci-1198444 v2 has been modified but not really improved, in my opinion.

Please consider the following suggestions:

Title is not appropriate. How about “mammary secretions” (including colostrum, transitional milk and milk)?

l. 29: components instead of "compositions"

l.29 : experimental theses instead “breeds”

l. 57: present, not presented

l. 66: please rephrase “resist well to environmental heat and diseases well”

l. 76: do the Authors mean experimental farm?

l. 84: mammary secretions?

l. 94: crossbred cannot be defined as breed (see in fact the discussion section). Use experimental groups or theses

l. 95: crossbred instead of “mixed breed”

l. 97-98: Please define the "in a separately conventional open-housing system": do you mean free stall? Were the experimental theses raised in different boxes?

l. 112: natural instead of “naturally”

Table 1: I agree with CB8 comment! However, the energy content should be expressed in Joule (J), according to the International System Units. Please use g/100g instead of %.  Moreover, ash data are in the wrong row.

l. 120-121: please rewrite the sentence

l. 132-133: That is not possible: please note that you sampled 15-20 ml colostrum (see l. 120)

l. 152: analyses instead of “analysis”

l. 158-166: as described above and in “Limitations”, data are not independent but are repeated measures on same does. According to text, data were not processed by repeated measure analysis of variance.

Table 2, 4 and 5 (Title): erase “analyzed by using a general linear model procedure” and just move mean values ± standard error of mean from the legend.

Table 4: g instead of “gm” (International System Unit)

Table 5: g/100 g instead of %

Table 5: please explain 7/6/5 9/6/5 reported for IgG

l. 213: shown instead of “demonstrated”

Table 6 : do not use “breed”, consistently with discussion

l. 238-239 : please rephrase

l. 242: please rephrase

l. 261-262: please rephrase

l. 272-273: as well-known, vit A is fat-soluble

l. 327: do you mean six times?

l. 352-361: erase 4.5 Limitations and move the explanation to M&M section

Author Response

Answer to reviewer comments

Title is not appropriate. How about “mammary secretions” (including colostrum, transitional milk and milk)?

Answer – Change as reviewer suggestion

  1. 29: components instead of "compositions"

Answer – Change as reviewer suggestion

l.29 : experimental theses instead “breeds”

Answer – Change as reviewer suggestion

  1. 57: present, not presented

Answer – Change as reviewer suggestion

  1. 66: please rephrase “resist well to environmental heat and diseases well”

Answer – Rephrase sentence to “resist well to environmental heat and diseases”

  1. 76: do the Authors mean experimental farm?

Answer – We added “this experimental farm”

  1. 84: mammary secretions?

Answer – Change as reviewer suggestion

  1. 94: crossbred cannot be defined as breed (see in fact the discussion section). Use experimental groups or theses

Answer – Change as reviewer suggestion

  1. 95: crossbred instead of “mixed breed”

Answer – Change as reviewer suggestion

  1. 97-98: Please define the "in a separately conventional open-housing system": do you mean free stall? Were the experimental theses raised in different boxes?

Answer – We added the different theses were raised in different boxes.

  1. 112: natural instead of “naturally”

Answer – Change as reviewer suggestion

Table 1: I agree with CB8 comment! However, the energy content should be expressed in Joule (J), according to the International System Units. Please use g/100g instead of %.  Moreover, ash data are in the wrong row.

Answer – Changed as reviewer suggestions

  1. 120-121: please rewrite the sentence

Answer – We added “milking”

  1. 132-133: That is not possible: please note that you sampled 15-20 ml colostrum (see l. 120)

Answer – We added more information about the amount of mammary secretion and procedure.  

  1. 152: analyses instead of “analysis”

Answer – Changed as reviewer suggestion

  1. 158-166: as described above and in “Limitations”, data are not independent but are repeated measures on same does. According to text, data were not processed by repeated measure analysis of variance.

Answer – The data already were analyzed by repeated measure analysis of variance of SAS program. In the SAS program, demand “proc mixed” as general linear mixed model procedure is repeated measures analysis of variance.

Table 2, 4 and 5 (Title): erase “analyzed by using a general linear model procedure” and just move mean values ± standard error of mean from the legend.

Answer – We removed “analyzed by using a general linear model procedure” and move “mean values ± standard error of mean”

Table 4: g instead of “gm” (International System Unit)

Answer – Change as reviewer suggestion 

Table 5: g/100 g instead of %

Answer – Change as reviewer suggestion 

Table 5: please explain 7/6/5 9/6/5 reported for IgG

Answer – Remove error of 7/6/5 9/6/5

  1. 213: shown instead of “demonstrated”

Answer – Change as reviewer suggestion

Table 6 : do not use “breed”, consistently with discussion

Answer – Change most of the “breed” to “group”

  1. 238-239 : please rephrase

Answer – We rephase the sentences as suggestions

  1. 242: please rephrase

Answer – We rephase the sentences as suggestion

  1. 261-262: please rephrase

Answer – We rephase the sentences as suggestion

  1. 272-273: as well-known, vit A is fat-soluble

Answer – We rephase the sentences as suggestion

  1. 327: do you mean six times?

Answer – Yes. We change to six times

  1. 352-361: erase 4.5 Limitations and move the explanation to M&M section

Answer – We moved the limitation to materials and methods and discussion sections.
